# Faster Time to Treatment Decision of Viscoelastic Coagulation Test Results through Improved Perception with the Animated Visual Clot: A Multicenter Comparative Eye-Tracking Study

**DOI:** 10.3390/diagnostics12051269

**Published:** 2022-05-19

**Authors:** Clara Castellucci, Julia Braun, Sadiq Said, Tadzio Raoul Roche, Christoph B. Nöthiger, Donat R. Spahn, David W. Tscholl, Samira Akbas

**Affiliations:** 1Institute of Anesthesiology, University of Zurich and University Hospital Zurich, 8091 Zurich, Switzerland; clara.castellucci@usz.ch (C.C.); sadiq.said@usz.ch (S.S.); tadzioraoul.roche@usz.ch (T.R.R.); christoph.noethiger@usz.ch (C.B.N.); donat.spahn@usz.ch (D.R.S.); samira.akbas@usz.ch (S.A.); 2Department of Epidemiology, Epidemiology, Biostatistics and Prevention Institute, University of Zurich, 8001 Zurich, Switzerland; julia.braun-gruebel@uzh.ch

**Keywords:** Visual Clot, blood coagulation, viscoelastic test, rotational thromboelastometry, point-of-care, eye-tracking, visual perception, avatar

## Abstract

As the interpretation of viscoelastic coagulation test results remains challenging, we created Visual Clot, an animated blood clot aiming to facilitate raw rotational thromboelastometry (ROTEM) parameters. This study investigated anesthesia personnel’s cognitive processing in managing simulated bleeding scenarios using eye-tracking technology. This multicenter, international, computer-based study across five large, central European hospitals included 35 participants with minimal to no prior experience interpreting viscoelastic test results. Using eye-tracking technology and an iPad tagged with quick response codes, we defined the time to treatment decision and the time on screen surface in seconds of correctly solved scenarios as our outcomes. The median time to treatment decision was 52 s for Visual Clot and 205 s for ROTEM (*p* < 0.0001). The probability of solving the scenario correctly was more than 8 times higher when using Visual Clot than when using ROTEM (Hazard ratio [HR] 8.54, 95% CI from 6.5 to 11.21; *p* < 0.0001). Out of 194 correctly answered scenarios of participants with the eye-tracker, 154 (79.4%) were solved with Visual Clot and 40 (20.6%) with ROTEM. Participants spent on average 30 s less looking at the screen surface with Visual Clot compared to ROTEM (Coefficient −30.74 s, 95% CI from −39.27 to −22.27; *p* < 0.0001). For a comparison of the two modalities in terms of information transfer, we calculated the percentage of time on the screen surface of the overall time to treatment decision, which with Visual Clot was 14 percentage points shorter than with ROTEM (Coefficient −14.55, 95% CI from −20.05 to −9.12; *p* < 0.0001). Visual Clot seems to improve perception and detection of coagulopathies and leads to earlier initiation of the appropriate treatment. In a high-pressure working environment such as the operating and the resuscitation room, correct and timely decisions regarding bleeding management may have a relevant impact on patients’ outcomes.

## 1. Introduction

Dynamic changes in a patient’s hemostasis during surgery often require quick and accurate diagnosis and individualized treatment [1]. Viscoelastic tests provide more rapid insight into the complex coagulation system than conventional laboratory tests [2,3,4]. Hence, such global hemostasis assays are implemented across multiple disciplines, including trauma, obstetrics, transplant and cardiac surgery [5,6,7,8,9]. The benefits of viscoelastic testing include a reduction in transfusion requirements, shorter hospital stay, fewer perioperative complications and lower treatment costs, which has led to a recommendation grade 1C in the European guidelines for managing trauma [1,10]. Despite these advantages and the widespread use of viscoelastic hemostatic tests, correct analysis of rotational thromboelastometry (ROTEM) remains challenging. To reduce the complexity of interpreting numerous hemostasis-relevant parameters and to display this information in a situation awareness-oriented manner, our team developed Visual Clot, an animated, real-time three-dimensional graphic representation of ROTEM parameters [11,12,13]. In previous computer-based studies comparing the use of Visual Clot with ROTEM, we found facilitated interpretation of hemostatic test results regardless of users’ expertise in viscoelastic resuscitation [14]. With Visual Clot, both experienced and inexperienced physicians improved their performance, selecting a more accurate pharmacological treatment faster, with higher diagnostic confidence and reduced perceived workload [12,14]. Furthermore, the anesthesia personnel considered Visual Clot as intuitive and easy to learn and declared it simplified decision making in situations of acute bleeding [15].

Eye-tracking devices provide valuable insights into the underlying thinking pathways among clinicians [16,17,18]. They provide information on spatial and temporal measurements, gaze coordinates, dwell time on areas of interest, as well as fixations and saccades [19]. Thus, eye-tracking may lead to a deeper understanding of the underlying analytical patterns that contribute to decision-making. As information transfer in this setting depends on visual attention, it may be used as a surrogate parameter for situation awareness and allows conclusions to be drawn [20,21]. Using eye-tracking technology, this study assessed the performance of anesthesia personnel in managing simulated, computer-based bleeding scenarios using either conventional ROTEM temograms or corresponding Visual Clot animations as viscoelastic test result representations. Examining the time on the screen surface and the time to treatment decision, we assessed the duration of cognitive processing and thus the complexity of the two mentioned viscoelastic display modalities. We hypothesized that Visual Clot enhances information perception and processing, advancing the development of user-centered, situation awareness-oriented visualization technologies.

## 2. Materials and Methods

This study analyzed further results and eye-tracking data from a recently published, prospective, multicenter, international, computer-based study examining the use of standard ROTEM and corresponding Visual Clot animations in simulated bleeding cases [12]. We collected the data between September 2020 and October 2020 from five tertiary care centers (the University Hospital Zurich and the Cantonal Hospital Winterthur in Switzerland, the University Hospital Frankfurt and the University Hospital Wuerzburg in Germany and the Hospital Clinic de Barcelona in Spain). The cantonal ethics committee in Zurich, Switzerland (Business Management System for Ethics Committees Number 2020-00906) and the centers in Germany and Spain assessed the study protocol and issued a declaration of no objection. Participation in this study was voluntary without monetary compensation. Each participant signed an informed consent form before commencing the study.

### 2.1. Description of the Visual Clot Technology

We developed Visual Clot by applying user-centered design principles to simplify the interpretation of viscoelastic tests and to support care providers in making better and more efficient decisions [13]. Based on reducing the overall complexity of the presentation by highlighting relevant information, the ROTEM data is continuously transformed and presented as an animated blood clot. The cut-off values used by the algorithm are based on the coagulation algorithm implemented in Zurich and are available in Appendix A (Appendix A) [22]. Visual Clot depicts the three main components of hemostasis—fibrin, plasmatic factors and platelets—as either sufficient or deficient. In accordance with the preset cutoff values, the information in Visual Clot is simplified and divided into three categories: Too low, normal or too high. In addition, hyperfibrinolysis and the presence of a heparin effect can be displayed in an animated fashion. The Visual Clot training video, available in Appendix A (Appendix A), explains each animated viscoelastic feature in further detail.

### 2.2. Study Design

This analysis is based on a recently published study, which included 35 participants with little to no experience in interpreting viscoelastic test results. At first, we equipped all participants, except for those in Barcelona due to logistical reasons and those wearing prescription glasses, with an eye-tracking device, calibrated it, and started recording. We used a Pupil Invisible mobile eye-tracking device (Pupil Labs, GmbH, Berlin, Germany) to capture visual fixations and saccades of participants observing Visual Clot and ROTEM scenarios. The eye-tracker recorded the position of the foveal vision on the screen 200 times per second (200 Hz) and with 0.5 degrees of visual angle accuracy [20]. Then, we defined specific areas of interest for the correctly solved ROTEM and Visual Clot scenarios, which were evaluated with the Pupil Labs proprietary software Pupil Player Version 5.5. We provide an example of the screen of both modalities with drawn areas of interest in Appendix A (Appendix A). Participants interpreted nine different bleeding cases, presented in randomized order, once as ROTEM and once as corresponding Visual Clot, resulting in 18 scenarios. After each scenario, the required pharmacotherapy had to be selected from a predefined multiple-choice questionnaire (Harvest Your Data, presented in the app iSurvey) on an iPad (Apple Inc., Cupertino, CA, USA) [23]. Figure 1 shows an exemplary bleeding case of combined plasmatic factors and fibrin deficiency and hyperfibrinolysis in both viscoelastic modalities. An overview of all cases with the respective correct answers according to the locally established and validated Zurich coagulation algorithm is available in Appendix A (Appendix A) [22].

### 2.3. Outcomes and Statistical Analyses

We defined the primary outcome of this study as the time to treatment decision in seconds, using an app-based timestamp to measure the time from the start of each scenario until the answers in the multiple-choice questionnaire were chosen. As the secondary outcome of this study, we examined the time on the screen surface of correctly solved scenarios.

For descriptive statistics, we show means with standard deviations and medians with interquartile ranges for continuous data and numbers and percentages for categorical data. With regard to the time to treatment, which is our initial outcome variable, Kaplan–Meier plots are used to show the probabilities for a correct decision in terms of the required pharmacotherapy and compare the two different modalities in an unadjusted way using the log-rank test. Despite not being ideal as this analysis ignores the dependencies between the repeated values from the same participant, it provides a first impression of the situation. We used mixed Cox regression models with a random effect for each participant to compare the two groups while adjusting for relevant covariates such as center, age, gender, job experience and the case due to repeated measurements.

In the second part of this report, we used only the times when the correct pharmacotherapy was chosen and analyzed the time on the screen surface. In addition, we calculated the percentage of the total time to treatment decision (between 0 and 100) spent looking at the screen surface (in contrast to reflection). The smaller this proportion is, the more time the participants are given to process the information and select the correct pharmacotherapy, which is desired from a clinical point of view. Using a mixed linear regression model, we compared these percentages between the two modalities and again adjusted the model for center, age, gender, job experience and the case. Further, we calculated a model for time on screen surface that only compares the time in seconds between the two modalities, not taking into account the total time needed to come to a decision nor the time participants did not look at the monitor (e.g., while reflecting).

For all statistical analyses, we used R Version 4.0.5 (R Foundation for Statistical Computing, Vienna, Austria) and considered a *p*-value < 0.05 to indicate statistical significance.

## 3. Results

### 3.1. Study and Participant Characteristics

We included 35 participants from five study centers. With 18 scenarios per participant (nine each for ROTEM and Visual Clot displaying the same bleeding case), we evaluated 315 scenarios per modality, collecting 630 results in total. Of the 351 (55.7%) correctly solved scenarios (69 (21.9%) using ROTEM, 282 (89.5%) using Visual Clot), usable eye-tracking data was available from 194 (55.3%) scenarios. In Barcelona, we were not able to collect eye-tracking data for logistical reasons. For six other participants, no eye-tracking data could be collected due to technical difficulties, such as wearing corrective glasses. Figure 2 presents a flowchart of all participants and analyzed scenarios. Table 1 shows the study and participant characteristics in detail.

### 3.2. Time to Treatment Decision

Regarding our primary outcome, there is very strong evidence for a shorter time to treatment decision for Visual Clot across all cases. The median time to treatment decision was 52 s for Visual Clot and 205 s for ROTEM, *p* < 0.0001. Figure 3 shows Kaplan–Meier curves for the time to treatment decision for both modalities.

Using mixed Cox models taking into account the dependencies of repeated measurements of the same participant, the probability of solving the scenario correctly was more than 8 times higher when using Visual Clot than when using ROTEM (Hazard ratio (HR) 8.54, 95% CI from 6.5 to 11.21; *p* < 0.0001).

In the detailed analysis, the various cases show different levels of difficulty based on the probability of successful completion. For instance, there was very strong evidence for an increased level of difficulty in bleeding cases with factor deficiency (HR 0.46, 95% CI from 0.3 to 0.72; *p* = 0.00058), hyperfibrinolysis (HR 0.37, 95% CI from 0.23 to 0.58; *p* < 0.0001) and factor plus fibrin deficiency and hyperfibrinolysis (HR 0.33, 95% CI from 0.21 to 0.52; *p* < 0.0001) compared to normal viscoelastic findings.

In regard to participants’ age, gender and the study center, we found no relevant evidence for an impact on the time to treatment decision. There was weak to moderate evidence for a shorter time to treatment decision with increasing professional experience (third year of residency compared to a fifth-year student: HR 2.64, 95% CI from 0.99 to 7.06, *p* = 0.05). We found no evidence for such an effect in the other experience categories.

### 3.3. Time on Screen Surface

Analyzing the eye-tracking data for all correctly solved scenarios, we found that when using Visual Clot, the anesthesia personnel correctly answered 154 out of 194 (79.4%) scenarios compared to 40 out of 194 (20.6%) scenarios when using standard ROTEM readings. Regarding the time on screen surface across both hemostatic display modalities, we found a median screen time of 26.5 s, which corresponded to 52.7% of the median time to treatment decision (i.e., 51.0 s).

There was a very high level of evidence that using Visual Clot resulted in a reduced time on screen surface (Coefficient −30.74 s, 95% CI from −39.27 to −22.27; *p* < 0.0001). In addition to the significantly shorter time on the screen surface, with a likewise significantly shorter time to treatment decision, it is particularly the percentage of the former that is relevant so that the conclusion can be drawn about how large the proportion of information perception and processing is in the total time to decision. In a mixed linear regression model adjusting for center, age, gender, job experience and case, the modality had a significant effect on the time on the surface by over 14 percentage points less for Visual Clot than ROTEM (Coefficient −14.55, 95% CI from −20.05 to −9.12; *p* < 0.0001). As an example, this means that if a participant spent 50% of their time on treatment decision looking at the screen using ROTEM, they only used 36% with Visual Clot. Hence, using Visual Clot shortened not only the time on the screen surface but also the percentage of the total time to treatment decision. In addition to the modality, particular cases also impacted the time spent on the screen surface. There is weak to moderate evidence for a prolonged time spent on screen surface for the bleeding case with factor, fibrin and platelet deficiency (difference for percentage spent on screen surface = 8.8, 95% CI from 0.42 to 17.25; *p* = 0.05) and the case with factor plus fibrin deficiency, hyperfibrinolysis (difference for percentage spent on screen surface = 11.18, 95% CI from 2 to 20.08; *p* = 0.02), which present the highest amount of pathological changes compared to the case with normal findings. There was no evidence of the influence of age and gender on the results.

Not only did the time to treatment decision decrease with more job experience, but also the percentage of time on the screen surface. There was a nonsignificant difference of 7 percentage points between the first (difference for percentage spent on screen surface = −3.02, 95% CI from −10.84 to 4.89; *p* = 0.52) and second (difference for percentage spent on screen surface = −10.9, 95% CI from −21.64 to −0.04; *p* = 0.12) year of residency. In Figure 4, we provide an analysis of the time on screen surface as boxplots.

## 4. Discussion

### 4.1. Principal Findings

This study evaluated data from 35 participants collected while interpreting viscoelastic test results of simulated bleeding cases, either using the animated Visual Clot or standard ROTEM temograms. We investigated the time to treatment decision and the time on screen surface for the correctly solved scenarios using eye-tracking data. The main findings comprise a significantly shorter time to treatment decision as well as both absolute and proportional shorter time on screen surface when using Visual Clot.

In the previous study, we found that participants were more likely to make the correct diagnosis, felt more confident in their therapeutic decision and reported less perceived workload with Visual Clot compared to ROTEM [14]. In this present study, we demonstrated that a viscoelastic test result presented with Visual Clot helped participants choose the correct pharmacotherapy significantly faster than with ROTEM (median 52 s versus 205 s, *p* < 0.0001). The measured data substantiates the perception of participants who stated that the interpretation of complex viscoelastic test results was faster with Visual Clot [15]. We consider this especially relevant in cases of severe hemorrhage, as not only the correct but also prompt treatment may strongly influence the patients’ outcome [24].

Since additional cognitive challenges in a high-pressure working environment such as the operating room or the emergency department lead to a decline in performance, efforts should be made to present medical information as simply as possible to reduce stress [25]. The shorter time to decide the respective treatment when using Visual Clot underlines the intuitiveness and ease of interpreting the animated viscoelastic test results, as previously shown by qualitative feedback provided by both viscoelastic-experienced and-inexperienced users [12,14,15].

Further, this study showed for the first time how professional experience influenced the performance of interpreting viscoelastic test results when using Visual Clot [14]. We found that increasing professional experience significantly decreased the time to treatment decision, suggesting that users with more work experience are generally more adept in perceiving and processing information quickly, even though they may be novices in viscoelastic resuscitation.

We sub-analyzed the eye-tracking data for all correctly managed scenarios as we found only them clinically relevant. Incorrectly solved scenarios often show a very short or excessively long time on the screen surface, indicating either complete incomprehension or long reflection, respectively. Almost 80% of the included scenarios were solved with Visual Clot, demonstrating again the significantly better performance [14].

Participants’ gaze rested on the screen surface for about half of the overall time to treatment decision without distinguishing between the two methods. We postulate that the other half was used for reflecting and answering the questions about the required pharmacotherapy.

Using mixed linear regression models, we demonstrated that the type of viscoelastic presentation significantly influenced the participants’ decision-making, as they looked at the screen surface for 30 s less with Visual Clot compared to ROTEM (*p* < 0.0001). In order to compare the two viscoelastic methods with regard to information transfer, we also calculated the proportion of the time on screen surface in the total time to treatment decision, which is significantly smaller by 14 percent (*p* < 0.0001) with Visual Clot. This indicates that information perception has been improved by the Visual Clot visualization technology compared to ROTEM. From a clinical point of view, it is desirable that the presentation of data is as simple as possible to perceive, allowing more time and cognitive capacity for treatment decisions.

This result suggests that with Visual Clot, the pathological values are detected faster and more easily and supports our efforts to present medical information in a simple and more intuitive form with the help of animations. Our study group has demonstrated with the Visual-Patient-avatar, a designated situation awareness tool for a patient’s vital signs, that animated visual presentations help the user perceive the information presented faster, in a more accessible and complete way [26,27,28].

### 4.2. Future Perspectives

The next step to evaluate the potential implementation in daily clinical practice is through a simulation study, where anesthesia personnel are required to solve critical bleeding scenarios by either using Visual Clot or ROTEM. This will allow further analysis of how the visual display aids clinicians in high-pressure situations. We currently understand Visual Clot as a supplement to numerical ROTEM data and temograms designed to obtain situation awareness rapidly, as it does not depict trends over time. A split-screen display of both modalities may offer the highest benefits in an actual clinical application. This would combine the higher diagnostic certainty as well as the presentation of the clot development whilst helping more inexperienced clinicians better interpret ROTEM.

### 4.3. Limitations and Strengths

There are several limitations to this study. As the data were collected as part of a previous study, we did not perform sample size calculations regarding our primary endpoint. Higher participant rates may influence our findings. Further, the participants were all from tertiary care hospitals in Central Europe and selected according to their clinical availability. Results may vary elsewhere in the world and in smaller medical facilities. However, we believe as all participants had little to no experience in interpreting viscoelastic test results that our results are transferable to smaller health care settings with lower exposure to viscoelastic resuscitation. Another inherent limitation is the interpretation of the eye-tracking data. Although a positive correlation between visual fixation and correct perception has been validated in the past, other influences on the perception may not be captured, such as peripheral vision and working memory [16,29]. Strengths of this study include a multicenter, international design, a balanced participant selection and the concept of within-subject comparisons, which minimizes interparticipant variabilities and thus may eliminate alternative explanations for our findings [30].

## 5. Conclusions

This computer-based simulation study underlines the potential of presenting medical information in a user-centered and situation awareness-oriented manner. The eye-tracking data revealed a faster perception of viscoelastic test results when using Visual Clot compared to using standard ROTEM tracings. In a clinical context, using Visual Clot may lead to improved perception, faster detection of underlying pathologies and earlier initiation of treatment. In a high-pressure working environment such as the operating and the resuscitation room, correct and timely decisions regarding bleeding management may have a relevant impact on patients’ outcomes. The results of this study emphasize the importance of further developing this technology. Therefore, we will evaluate the Visual Clot technology in a planned high-fidelity simulation study, aiming to further improve patient outcomes through possible future implementation in clinical practice.

## Figures and Tables

**Figure 1 diagnostics-12-01269-f001:**
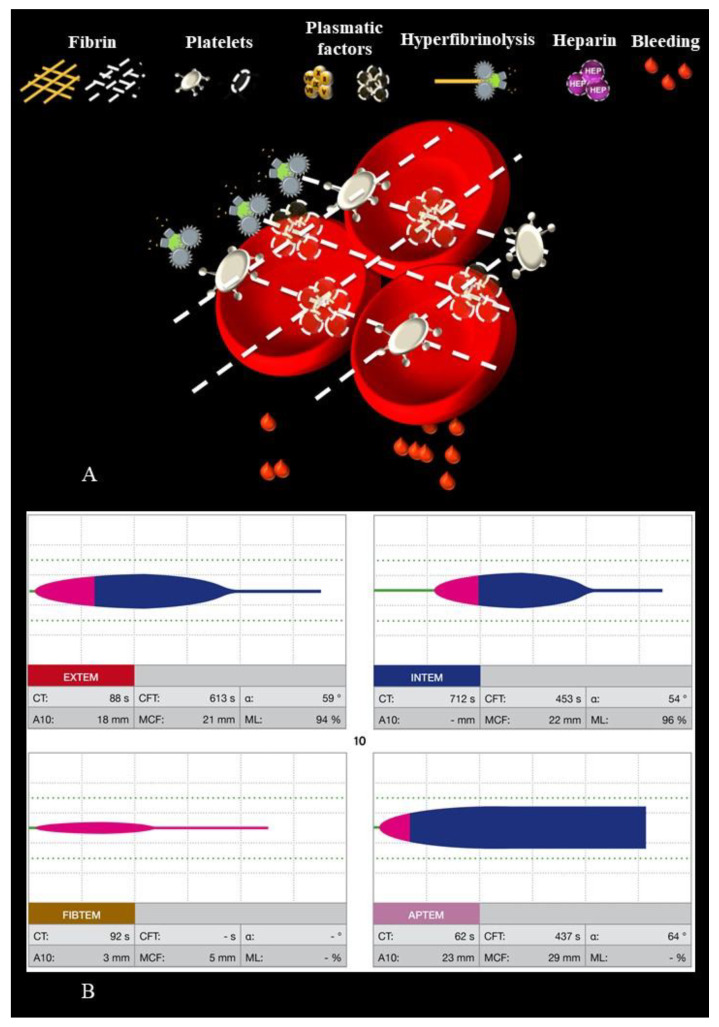
Combined plasmatic factors and fibrin deficiency, hyperfibrinolysis shown as Visual Clot and ROTEM. (**A**): Visual Clot with fibrin deficiency displayed as dashed lines, deficiency of plasmatic factors displayed as dashed lines and hyperfibrinolysis. (**B**): corresponding ROTEM with EXTEM, INTEM, FIBTEM and APTEM. CT = Clotting Time; CFT = Clot Formation Time; MCF = Maximum Clot Firmness; ML = Maximum Lysis.

**Figure 2 diagnostics-12-01269-f002:**
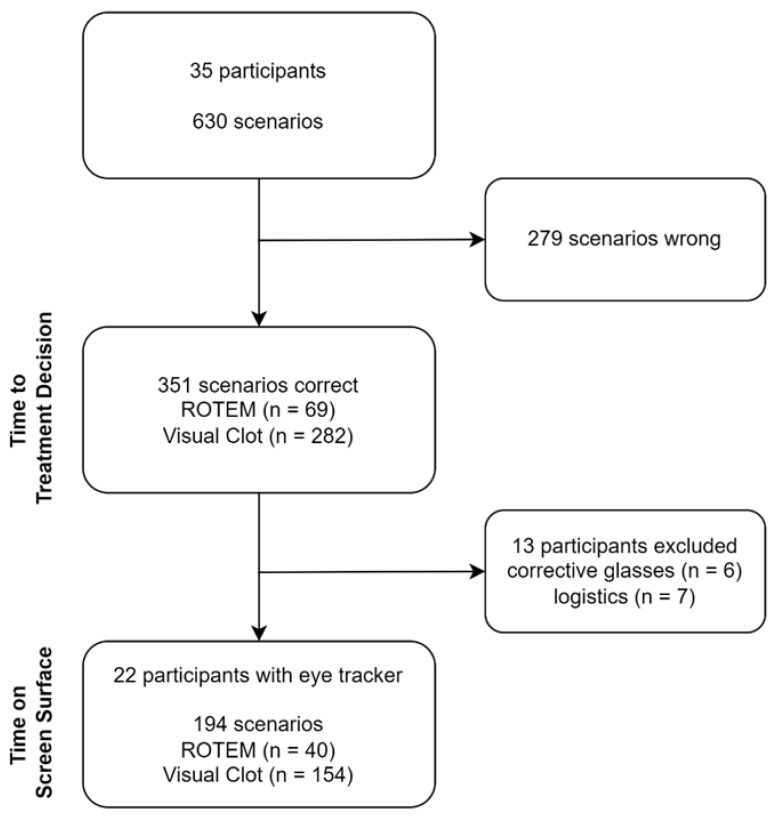
Flowchart presenting the total participant (n = 35) and scenario numbers (n = 630). Time to treatment decision is based on correctly answered scenarios (n = 351) of all participants. Time on screen surface is based on correctly answered scenarios (n = 351) of all participants. Time on screen surface is based on correctly answered scenarios (n = 194) of participants wearing the eye-tracking device (n = 22). ROTEM = rotational thromboelastometry.

**Figure 3 diagnostics-12-01269-f003:**
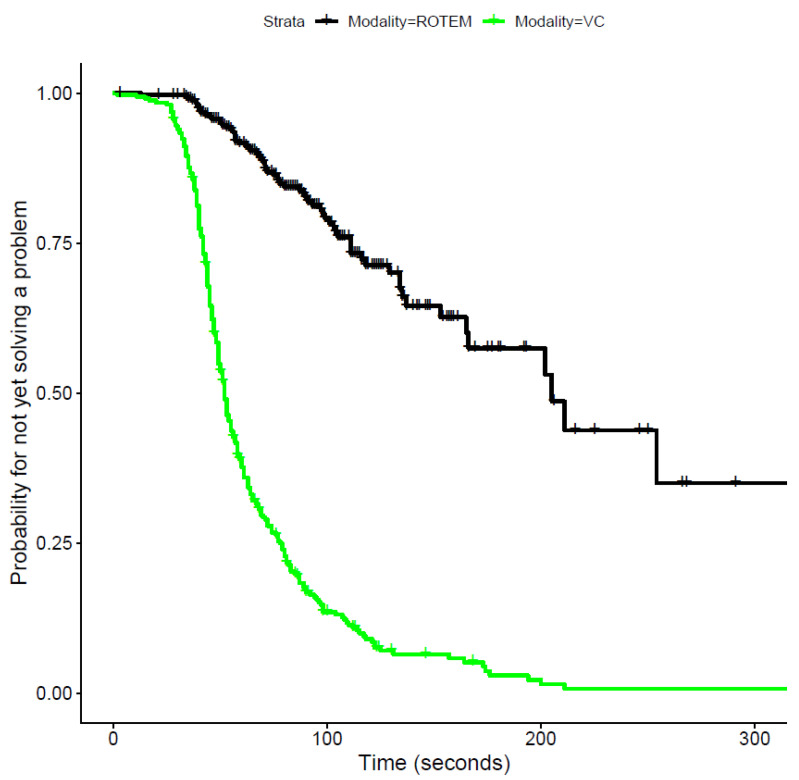
Kaplan–Meier curves for the time to treatment decision regarding ROTEM and Visual Clot (*p* < 0.0001). The *y* axis describes the probability of not yet solving a problem. The faster this probability decreases, the faster the scenario can be solved. ROTEM = rotational thromboelastometry; VC = Visual Clot.

**Figure 4 diagnostics-12-01269-f004:**
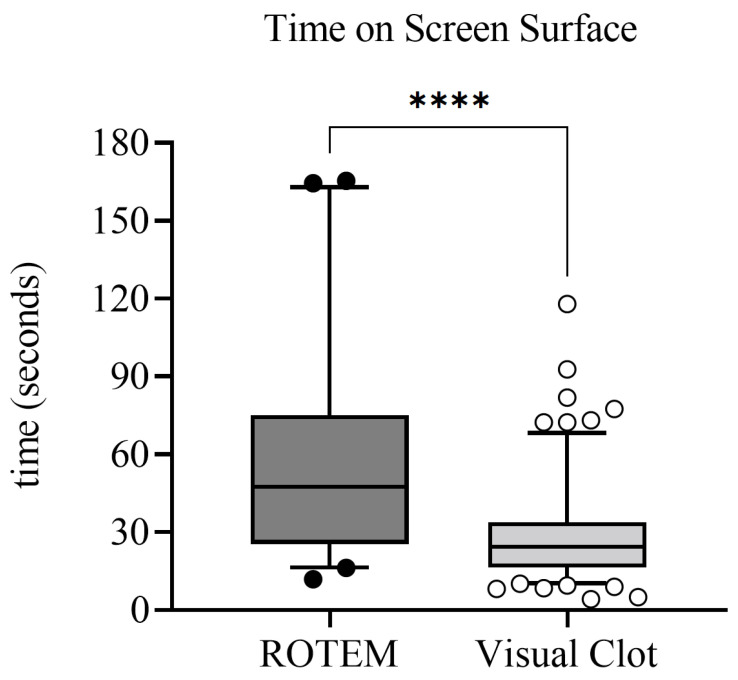
Boxplots representing the time spent on screen surface for either correctly solved ROTEM (n = 40) or Visual Clot (n = 154) scenarios. The box depicts the first and third quartiles, with the line indicating the median. The whiskers represent the 5th and 95th percentile, and the dots any results above or below the range. The asterisks indicate the level of significance from the mixed linear regression model, where it was taken into account that times from the same individual are not independent (*p* < 0.0001). ROTEM = rotational thromboelastometry.

**Table 1 diagnostics-12-01269-t001:** Study and participant characteristics in detail. We present the data as numbers (%) or median (IQR) interquartile range (range). ROTEM = rotational thromboelastometry.

Characteristic	Number
Participants, n (%)	35 (100)
University Hospital Zurich	7 (20)
Cantonal Hospital Winterthur	7 (20)
University Hospital Frankfurt	7 (20)
University Hospital Wuerzburg	7 (20)
Hospital Clinic de Barcelona	7 (20
Participants with successful eye-tracking recording, n (%)	22 (63)
University Hospital Zurich	6 (17)
Cantonal Hospital Winterthur	5 (14)
University Hospital Frankfurt	7 (20)
University Hospital Wuerzburg	4 (11)
Hospital Clinic de Barcelona	0 (0)
Gender, n (%)	35 (100)
Female	17 (49)
Male	18 (51)
Age (years), median (IQR, range)	28 (25–32, 24–36)
Job Experience, n (%)	35 (100)
Fifth-year medical student	3 (8)
Sixth-year medical student	10 (29)
First-year resident	19 (54)
Second-year resident	2 (6)
Third-year resident	1 (3)
Previously used ROTEM, n (%)	35 (100)
Yes	9 (26)
No	26 (74)
Previously used Visual Clot, n (%)	35 (100)
Yes	1 (3)
No	34 (97)

## Data Availability

The datasets used and/or analyzed during the current study are available from the corresponding author on reasonable request.

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
