# Peer review of "Faster Time to Treatment Decision of Viscoelastic Coagulation Test Results through Improved Perception with the Animated Visual Clot: A Multicenter Comparative Eye-Tracking Study"

_diagnostics, 2022, doi:10.3390/diagnostics12051269_

Round 1
Reviewer 1 Report
Major issues:
- The authors should add p-value in Figure 3.
- The authors should denote what kind of statistical analysis method they use to derive the p-value in Figure 4.
- Figure 4, make the fonts in x-, and y- axis larger.
Author Response
Dear Reviewer 1
Thank you very much for your valuable feedback and suggestions. Accordingly, we revised the manuscript to mend weaknesses and hope that we reached the demanded improvements.
1: The authors should add p-value in Figure 3
Answer: thank you very much for this information. We have added the p-value.
2: The authors should denote what kind of statistical analysis method they use to derive the p-value in Figure 4.
Answer: the p-value in figure 4 was calculated using a mixed linear regression. We have added the information to the description of the figure.
3: Figure 4, make the fonts in x-, and y- axis larger.
Answer: thank you very much for this advice. Readability is very important and we have adjusted the font size accordingly.
Reviewer 2 Report
The manuscript entitled "Faster Time to Treatment Decision of Viscoelastic Coagulation Test Results through Improved Perception with the Animated Visual Clot: A Multicenter Comparative Eye-Tracking Study" presents the advantages of using Visual Clot with impact on the improvement of the perception and detection of coagulopathies.
The manuscript is well prepared, and I recommend its acceptance for publication after considering the following minor revisions:
Carefully check the entire manuscript for typos;
Organize the figures and figure captions (for example, the caption for Figure 3 is before this differently from the others);
For the readers' convenience, mention in the manuscript that A1-A4 are supplementary files;
Consider inserting „Future perspectives” and highlight the main advantages of using Virtual Clot in connection to ROTEM, as well as the possibilities to remove the actual limitations.
Author Response
Dear Reviewer 2
Thank you very much for your valuable feedback and input. We revised the manuscript accordingly to mend weaknesses and hope that we reached the demanded improvements.
1: Carefully check the entire manuscript for typos
Answer: Thank you very much. we have carefully revised the manuscript and corrected as many typing errors as possible.
2: Organize the figures and figure captions (for example, the caption for Figure 3 is before this differently from the others)
Answer: The figure captions should now be in the correct order after the revision.
3: For the readers' convenience, mention in the manuscript that A1-A4 are supplementary files
Answer: Thank you also for this important tip. We have added that A1-A4 are supplementary files.
4: Consider inserting „Future perspectives” and highlight the main advantages of using Virtual Clot in connection to ROTEM, as well as the possibilities to remove the actual limitations.
Answer: We really appreciate your suggestion to change the section 'limitations of visual clot' to 'future perspectives'. We also think that the visual clot is still in the development phase and therefore we would like to talk less about limitations and much more about future potential.